# Is the Surgical Drainage Mandatory for Leak after Sleeve Gastrectomy?

**DOI:** 10.3390/jcm12041376

**Published:** 2023-02-09

**Authors:** Marius Nedelcu, Thierry Manos, Patrick Noel, Marc Danan, Viola Zulian, Ramon Vilallonga, Anamaria Nedelcu, Sergio Carandina

**Affiliations:** 1ELSAN, Clinique Saint Michel, Centre Chirurgical de l’Óbesite, 83000 Toulon, France; 2ELSAN, Clinique Bouchard, 13006 Marseille, France; 3Emirates Specialty Hospital, Dubai Healthcare City, Dubai 505240, United Arab Emirates; 4Mediclinic Airport Road Hospital, Abu Dhabi 48481, United Arab Emirates; 5Endocrine, Metabolic and Bariatric Unit, General Surgery Department, Hospital Vall d’Hebron, 08035 Barcelona, Spain; 6Surgery Department, Universitat Autònoma de Barcelona, 08193 Barcelona, Spain

**Keywords:** sleeve gastrectomy, endoscopy, surgical drainage, double pigtail, septotomy

## Abstract

Introduction: Despite the unanimous acknowledgement of the laparoscopic sleeve gastrectomy (LSG) worldwide, the leak remains its deficiency. For the last decade, the surgical treatment was practically considered mandatory for almost any collection following LSG. The aim of this study is to evaluate the need for surgical drainage for leak following LSG. Methods: All consecutive patients having gone through LSG from January 2017 to December 2020 were enrolled in our study. Once the demographic data and the leak history were registered, we analyzed the outcome of the surgical or endoscopic drainage, the characteristics of the endoscopic treatment, and the evolution to complete healing. Results: A total of 1249 patients underwent LSG and the leak occurred in 11 cases (0.9%). There were 10 women with a mean age of 47.8 years (27–63). The surgical drainage was performed for three patients and the rest of the eight patients underwent primary endoscopic treatment. The endoscopic treatment was represented with pigtails for seven cases and septotomy with balloon dilation for four cases. In two out of these four cases, the septotomy was anticipated by the use of a nasocavitary drain for 2 weeks. The average number of endoscopic procedures was 3.2 (range 2–6). The leaks achieved complete healing after an average duration of 4.8 months (range 1–9 months). No mortality was recorded for a leak. Conclusions: The treatment of the gastric leak must be tailored to each patient. Although there is still no consensus for the endoscopic drainage of leaks after LSG, the surgical approach can be avoided in up to 72%. The benefits of pigtails and nasocavitary drains followed by endoscopic septotomy are undeniable, and they should be included in the armamentarium of any bariatric center.

## 1. Introduction

Morbid obesity’s outgrowing prevalence is a mondial health problem. Its best treatment nowadays is surgery, generating an explosion in the number of bariatric procedures worldwide—Laparoscopic Sleeve Gastrectomy (LSG) in particular [1,2]. LSG offers a better life quality comparing to the gastric banding [3] and a reduced risk of long-term complications compared to RYGBP which has at least one complication in more than 35% of patients within the 10-year follow-up period [4]. The staple line gastric leak (GL) is unanimously accepted as the most feared complication of LSG, due to its difficult healing process, especially when the treatment approach is non standardized, with a currently reported rate of 0.3 to 0.4% [5]. Studies occasionally incriminated the learning curve for different bariatric procedures as essential in determining the postoperative complication rates [6,7].

The clinical presentation of the leak after LSG is variable, ranging from low-symptomatic patients with minimal left shoulder pain to patients presenting symptoms of sepsis with generalized peritonitis. If for the hemodynamically unstable patients the surgical reintervention is mandatory, for the low symptomatic patients there is no urgency, and the therapeutic decision should be discussed. At the early beginnings of LSG, early surgical reintervention with drainage was recommended for any collection close to the staple line [8]. After the initial phase, some progress was done towards the conservative treatment of the leak [9,10], with a small category of patients being treated without a surgical reintervention. The radiological drainage is referred to as another option to treat the collections following LSG [11,12,13]. Nowadays, the endoscopy has progressed not only towards becoming the definitive treatment for the leak but also to being used for the initial drainage in the acute phase, and the term of endoscopic internal drainage is very well established in many practices [14]. Concerning the definitive treatment, after an initial enthusiasm for stents, the endoscopic approach evolved, including septotomy with a balloon dilatation and pigtails insertions in the current management. The purpose of the current manuscript is to evaluate the rate of surgical drainage and the outcomes of an exclusively endoscopic approach for the treatment of a leak following LSG.

## 2. Materials and Methods

All consecutive patients who underwent LSG in Saint Michel Private Hospital (Toulon, France) between January 2017 and December 2020 were included in the current study. All files of the patients diagnosed with leak in the postoperative period were carefully reviewed and their outcome analyzed retrospectively. After collecting the demographic data (age, gender BMI), the symptomatology was analyzed along with the radiological studies. Patient’s hemodynamic status directed the treatment approach in performing the drainage of the collection through laparoscopy or with endoscopy alone. The parenteral nutrition, triple antibiotics, and fluid management were systematically associated. No feeding jejunostomy was used. In case of surgical drainage, a systematic endoscopy was performed in the first two weeks after surgery. In case of a patient with no hemodynamic instability, the endoscopic approach was preferred, and the endoscopic drainage was realized using pigtail drains when the orifice was smaller than 10 mm or with a nasocavitary drain in the case of larger orifices. When the patient was unstable and needing the control of a severe infection, the first step was the laparoscopic lavage with drainage of the peritoneal cavity, which were realized in an emergency setting. The algorithm of the endoscopic treatment (Figure 1) was completed with endoscopic septotomy associated with balloon dilation or pigtail insertion, depending on the leak size, time of diagnosis, and associated stenosis. We have also analyzed the number of endoscopic sessions, the duration of treatment, and the healing rate for the endoscopic treatment. During this approach, parenteral nutrition was administered for 2 weeks with repetitive studies being performed between 4 and 6 weeks. The entire healing process was protected through the administration of a double dose of Proton Pump Inhibitors (PPIs).

All procedures carried out in studies on human participants complied with the ethical standards of the national research committee and with the 1964 Helsinki declaration and its later amendments or comparable ethical standards. All individual participants included in the study signed an informed consent. The ethical board of our institution gave its approval for the study.

### Statistical Analysis

Continuous demographic variables were communicated as mean ± standard deviatio, and range; categorical variables as well as complications were analyzed as numbers and percentages. Continuous outcome variables were generally expressed as mean ± standard deviation, and range. We used descriptive statistics (simple counts and mean values) to declare the complications and adverse effects.

## 3. Results

A total of 1249 patients underwent LSG between January 2017 and December 2020. The leak was recorded in 11 cases (0.9%). There were 10 women with mean age 47.8 years (27–63). All patients were operated on by two surgeons (MN and SC). Leak rate distribution per year is illustrated in Figure 2. Regarding the time diagnosis of any leak we have recorded, they are as follows: two cases of early, acute leak (<4 days); five cases of intermediate leak (4–7 days), and four cases of late leak (>7 days).

The surgical drainage was necessary in only three patients (27.3%) for the following reasons: one hemodynamically unstable patient, one patient with generalized peritonitis, and one patient with no endoscopic treatment availability. For the rest of the eight patients, they underwent primary endoscopic drainage using pigtails in six cases and nasocavitary drainage in two cases. The radiological drainage was never used in our experience. For the six patients, one double pigtail drain (7 French in diameter and 4 or 5 cm in length) was inserted both as the primary drainage and the first line of treatment. Four to five weeks later, the endoscopic control revealed complete healing in two patients, with the necessity of a pigtail drain replacement for the rest of the four patients, among whom the leak was completely healed and the drain removed 6 weeks later during the third endoscopy. In two cases, the orifice of leak identified with endoscopy was large with an easy passage of the endoscope outside of the gastric lumen. The decision was to place a nasocavitary drain with the purpose of performing repeated (3–5 times a day) lavages of the cavity for 2 weeks in order to obtain clean tissue. The treatment was completed with septotomy associated with systematic balloon dilatation, and, in one patient, an additional pigtail was inserted in a secondary trajectory. The secondary trajectory is defined as a trajectory departing from the former peritoneal abscess cavity. For the rest of three patients that underwent initial surgical drainage, the definitive treatment consisted in repetitive insertions of pigtails (one case) or endoscopic septotomy with balloon dilatation (two cases). For all three cases the surgical drainage was removed in the first 72 h after the endoscopic treatment.

The average number of endoscopic procedures was 3.2 (range 2–6). One patient was transferred to the thoracic surgical department for a pulmonary abscess associated with hemoptysis and treated with interventional radiology. The leaks achieved complete healing after an average duration of 4.8 months (range 1–9 months). No mortality was recorded.

## 4. Discussion

Gastric leak (GL) is estimated to be the most serious complication of this procedure due to multifactorial alterations of wound healing near the gastroesophageal junction, such as increased pressure, stricture formation and a too narrow sleeve, mismatched staple height and tissue thickness, vascular supply, and energy sources, among others. Even if the GL prevalence has decreased steadily, it remains highly variable between 0 and 18% [7,15,16]. LSG remains an unpredictable procedure with no clearly identified risk factors for leak.

When faced with a GL, the surgeon must consider all options to confront it adequately. After leak diagnosis, all unstable patients need drainage of the collection and lavage of the peritoneal cavity with laparoscopy. Instead, for stable patients, the approach could be discussed, and it has evolved over time. In the early beginning, the systematic surgical drainage was advocated. Later on, a conservative treatment with antibiotic therapy, and no drainage was attempted for certain types of leaks [10]. Other teams have gained important experience with the drainage of collections through interventional radiology. Nowadays, in our experience, the Endoscopic Internal Drainage represents the first intention treatment to manage collections close to the staple line for the hemodynamically stable patients.

Bariatric surgery made tremendous progress during the last decade. We have largely adopted the ERABS (enhanced recovery after bariatric surgery) protocols with the abandonment of the nasogastric tube, surgical drain, or systematic urinary catheter [17]. A similar progress should be done in treating the complications following bariatric surgery. The dogma to systematically adopt a surgical drainage for any abdominal collection close to the staple line should be reevaluated. In our experience, between January 2017 and December 2020, 8 patients out of 11 (72.7%) had a drainage with endoscopy exclusively. Avoiding the surgical drainage, the risk to develop gastro-cutaneous fistula is greatly decreased. Instead, blood and electrolyte imbalance restoration, alimentary tract resting, optimal nutrition launching, and the sepsis management should remain the absolute priorities. We always needed to proceed with the endoscopic exploration of the gastric area afterwards, to assess for different methods of endoscopic treatment.

The efficiency of the endoscopic treatment for the leak after LSG dominates in the standardization of this type of management; therefore, we recommend all tertiary bariatric centers to implement a specific algorithm which should to be re-evaluated and updated every 3 to 5 years. Our previously published algorithm for leak treatment [18] based on the size of the fistulous site permitted us to diminish the number of endoscopic interventions, but stents have their limitations: a poor quality of life and the potential of life-threatening complications (e.g., aorto-esophageal fistula) [19]. Subsequently, the concept of septotomy was developed, initially for the treatment of chronic leaks with an orifice measuring over 10 mm large and allowing the passage of the endoscope into the cavity [19,20]. Furthermore, along with the septotomy, a pneumatic dilation of 20 mm of the sleeve was performed to improve the quality of the drainage. Here are the ideas we would like to stress with our currently updated algorithm:The endoscopic septotomy has replaced the stents, especially in the case of late leaks.We need to acknowledge the importance of time between the initial operation and the leak diagnosis; the septotomy can be performed once the chronical inflammatory tissue is clearly identified, preferring a nasocavitary drainage in the acute phase.We did not find any utility of the OVESCO system in our experience.The concept of secondary trajectory with the pigtail insertion outside the lumen cavity was conceived and probably is described in the literature for the first time.

When approaching a leak during the acute phase, if the patient is unstable hemodinamically or the endoscopic approach is unavailable, we drain with laparoscopy. In the other case scenario, we drain with endoscopy, during the pigtail insertion for the small orifices or while placing a naso-cavitary drain for large staple line dehiscence, in which case the drain is also used for regular wash outs for the next two weeks. The endoscopy also enables a leak cartography, facilitating the placement of double pigtails in the case of secondary orifices. The combination of septotomy plus dilation allowed us to successfully treat all large leaks in our series. In two patients, two secondary septotomies were necessary to complete the first septotomy which were partial due to an intense inflammatory process. We preferred to realize a two-step septotomy in these situations when the inflammation is very important and the tissues are not solid.

Concerning the association between leak and stenosis, we consider endoscopic pneumatic dilatation a valid solution, and we use it systematically. We perform the procedure with an achalasia balloon (Rigiflex^®^ balloon 30 mm) over a stainless steel or super-stiff guide wire, progressively dilating with stepwise increments in dilation pressure from 15 to 25 psi. Our best results are seen after aggressive inflation under radiological guidance, and, at the end of the procedure, we can easily confirm the correction of the gastric tube’s axis.

Another form of endoscopic internal drainage described in the literature representing a progress in bariatric endoscopy is the endoscopic vacuum therapy [21,22]. Even with an increased number of endoscopic evaluations, this therapy shares the same concept as pigtails or nasocavitary drains, assuring an internal drainage to the stomach and avoiding the gastro-cutaneous fistula. Hodge et al. [23] reported the harmful effect of drain in bariatric surgery after the analysis of 148,260 patients, the data from 2017 Metabolic and Bariatric Surgery Accreditation and Quality Improvement Program Participant Use File. Similarly, the external surgical drainage of a leak cavity following LSG will keep open the trajectory between the stomach and the skin, increasing the risk to develop a gastro-cutaneous fistula.

In our opinion, the collection drainage of the GL after LSG must be tailored to the hemodynamic status of the patient and to the experience and the availability of the endoscopic approach. Initially, the endoscopic drainage has been proposed only for patients with a “small” leak and minimal collection. Increasing our experience with the introduction of the endoscopic septotomy, the endoscopy became the first line of treatment using nasocavitary drains, even for patients with large orifices and important collections. This way, we avoid the risk of a gastro cutaneous fistula created by the surgical or radiological drainage.

Even if the main limitation of our study is represented by the reduced number of patients with GL following LSG, we believe that it provides a reliable picture of the modifications of the dogma imposing systematic reintervention for any collection close to the staple line. The endoscopic internal drainage represents the first-line treatment in our experience, and its success depends on two main factors: the availability of the endoscopic approach in close collaboration with the surgeon and the high expertise of the endoscopic team.

## 5. Conclusions

The treatment of the gastric leak must be tailored to each patient. Although there is still no consensus for the type of drainage of perigastric collections, the surgical approach was avoided in our experience in up to 72%. Avoiding the surgical drainage, the risk to develop gastro-cutaneous fistula is greatly decreased. The benefits of pigtails and nasocavitary drains followed by endoscopic septotomy are undeniable, and they should be included in the armamentarium of any bariatric center. Accordingly, it was possible to achieve an acceptable healing rate of leak following LSG with an improved quality of life and to reduce the number of endoscopic sessions and adverse events associated with the use of stents.

## Figures and Tables

**Figure 1 jcm-12-01376-f001:**
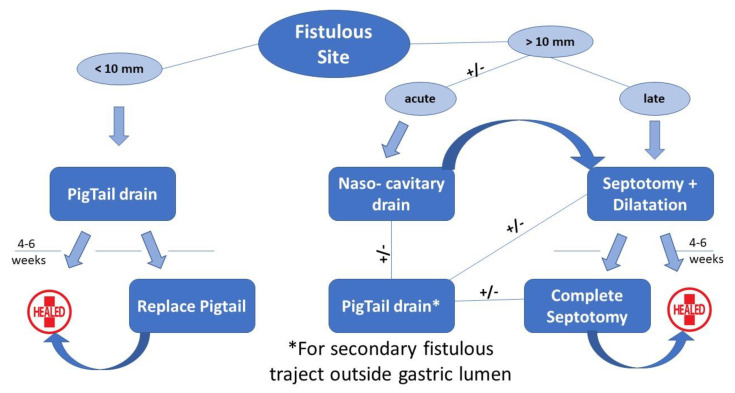
Algorithm of endoscopic treatment of leaks after LSG.

**Figure 2 jcm-12-01376-f002:**
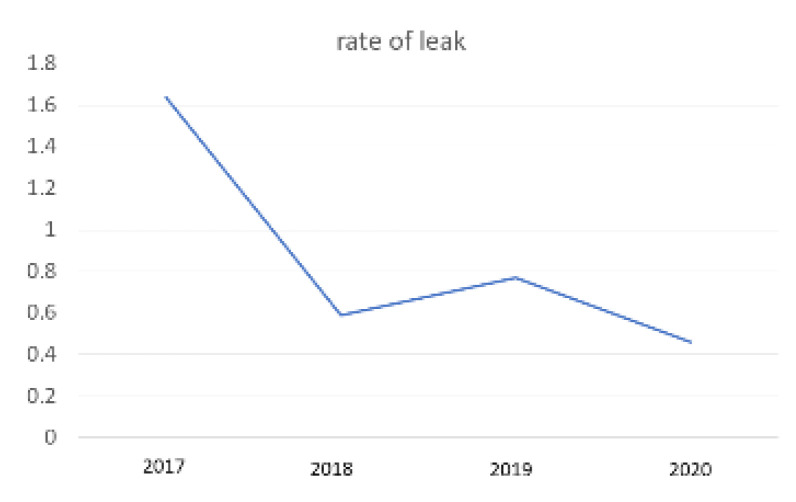
Leak rate—yearly distribution.

## Data Availability

The data was included in SOFFCO registry—French society of Bariatric Surgery.

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
