# Peer review of "Is the Surgical Drainage Mandatory for Leak after Sleeve Gastrectomy?"

_jcm, 2023, doi:10.3390/jcm12041376_

Round 1
Reviewer 1 Report
Dear Authors, congratulations for both the results and the management of fistula after gastric sleeve. The algorithm you indicated based upon your high experoence is very useful for practice.
Author Response
Dear Authors, congratulations for both the results and the management of fistula after gastric sleeve. The algorithm you indicated based upon your high experience is very useful for practice.
Thank you very much for your positive remarks. Consecutively to the other reviewers comments we have done the appropriate modifications according to our experience and convictions. We are convinced that by the modifications done to the manuscript, we have improved the quality of our paper.

Reviewer 2 Report
The authors present a single-centre, series of patients who underwent laparoscopic sleeve gastrectmy. The novelty of the contribution was represented by the fact that the managment of gastric leak, that remains the most common complication, was investigated. Tere are several limitations in this study: (i) it was conducted in a single midle-volume center; (ii) the study is retrospective and not prospective; and (iii) the number of cases is relatively low insufficient, and the results may be biased. Perioperative complications and outcomes seem good. The authors conclude that conservative methods in case of GL after LSG could be the first line treatment. The study offers some interesting insights.
Author Response
The authors present a single-centre, series of patients who underwent laparoscopic sleeve gastrectomy. The novelty of the contribution was represented by the fact that the management of gastric leak, that remains the most common complication, was investigated.
We thank the reviewers for their fair and very constructive feedback. We have done the appropriate modifications according to our experience and convictions. We are convinced that by the modifications done to the manuscript according to your suggestions we have highly improved the quality of our paper.
There are several limitations in this study:
- it was conducted in a single middle-volume center;
Thank you for your comment. Our center has a volume of around 400 patients/year – in the current study we have discussed only the sleeve. In our region – PACA – South of France it is the busiest bariatric private practice. We agree that with other countries this could be interpreted like a middle volume center, even if there is no classification of middle or high-level volume center.
- the study is retrospective and not prospective;
This was a retrospective study with a prospective database. According to your suggestion, we have included in the revised form of the manuscript the term “retrospective” in the methods in order to avoid any confusion.
- the number of cases is relatively low insufficient, and the results may be biased. Perioperative complications and outcomes seem good. The authors conclude that conservative methods in case of GL after LSG could be the first line treatment. The study offers some interesting insights.
The number of leaks has dramatically decreased in the last years to a percentage of less than 1 %, which is extremely positive for the patients and surgeons, but with this progress it will become even more difficult to analyze the factors involved in the occurrence of leak. Thank you very much for remark regarding the outcomes. Indeed our manuscripts emphasizes the change of a dogma that we have used for many years that “every patient with a leak need a surgical drainage”.

Reviewer 3 Report
At Line 140 and 214-215, when speaking about the correlation between surgical drain and gastro cutaneous fistula, there is no reference. The authors should also circumstantiate more that affirmation.
Author Response
We thank the reviewers for their fair and very constructive feedback. We have done the appropriate modifications according to our experience and convictions. We are convinced that by the modifications done to the manuscript according to your suggestions we have highly improved the quality of our paper.
At Line 140 and 214-215, when speaking about the correlation between surgical drain and gastro cutaneous fistula, there is no reference. The authors should also circumstantiate more that affirmation.
Thank you for the remark. According to your suggestion we have included in the revised form of the manuscript, the following paragraph and reference: “Hodge et al. [23] reported the harmful effect of drain in bariatric surgery after the analysis of 148 260 patients, the data from 2017 Metabolic and Bariatric Surgery Accreditation and Quality Improvement Program Participant Use File. Similarly, the external surgical drainage of leak cavity following LSG will keep open the trajectory between the stomach and the skin, increasing the risk to develop a gastro-cutaneous fistula.”
Gray EC, Dawoud F, Janelle M, Hodge M. Drain Placement During Bariatric Surgery, Helpful or Harmful? Am Surg. 2020 Aug;86(8):971-975

Reviewer 4 Report
Dear author,
You have a very interesting title which is area of concern for many bariatric surgeons.
But i have the following comments:
1- The title Is the Surgical Drainage Mandatory For Leak after Sleeve Gas-trectomy.
you should correct gas-trectomy to be one word.
2-In abstract section the results are much more and needed to be less.
3-The time of post operative leak if its acute or delayed as it will affect your management and results?
4-The possible causes of leak in your patients ( stenosis or other causes and its percentage)
5-The site and size of leak
6-limitation of your study
Best wishes
Author Response
Dear author,
You have a very interesting title which is area of concern for many bariatric surgeons.
But I have the following comments:
1- The title Is the Surgical Drainage Mandatory For Leak after Sleeve Gas-trectomy.
you should correct gas-trectomy to be one word.
Thank you very much for your remark. The word was corrected. This error occurred with the submissions and automat editing of the special template of JCM.
2-In abstract section the results are much more and needed to be less.
This section was reduced and modified according to your suggestion.
3-The time of post operative leak if its acute or delayed as it will affect your management and results?
Thank you very much for your question. Accordingly, the following paragraph was included in the revised form of the manuscript: “Regarding the time diagnosis of leak we have recorded: 2 cases of early, acute leak (< 4 days); 5 cases of intermediate leak (4-7 days) and 4 cases of late leak (> 7 days)”
4-The possible causes of leak in your patients ( stenosis or other causes and its percentage).
We completely agree with you that in the literature, we have only limited information about the causes of the leak following sleeve. Also, in our current experience in only 2 cases the leak could be explained by the history of migrated gastric band with a large area of ischemic, necrotic tissue. For all the other cases no clear risk factor for leak was identified.
For all 4 patients with septotomy a systematic associated balloon dilatation was performed in the presence or not of a functional stenosis at the level of the angulus.
5-The site and size of leak
All the leak diagnosed in the current were localized exclusively in the upper of the gastric tube, at the level of the Hiss angle. Moreover before or after the study period all the leaks diagnosed in our center have had the same localization.
Regarding the size of leak: 4 cases have the orifice of the leak larger than the endoscope; there is already this information in the initial form of the manuscript:
Lines 124-126: “In 2 cases, the orifice of leak identified by endoscopy was large with an easy passage of the endoscope outside of the gastric lumen.”
Lines 131-133: “. For the rest of 3 patients that underwent initial surgical drainage, the definitive treatment consisted in repetitive insertion of pigtails (1 case) or endoscopic septotomy with balloon dilatation (2 cases).” In all cases in which the septotomy was performed are larger than 10 mm and in cases when the pigtail was inserted for a primary trajectory are < 10 mm.
6-limitation of your study
Some limitation of the study could be interpreted that the study it is a single center and the number of cases included are limited, but nowadays with dramatically decrease of leak rate, the number of leaks following LSG have diminished.
Best wishes
We thank you for the fair and very constructive feedback. We have done the appropriate modifications according to our experience and convictions. We are convinced that by the modifications done to the manuscript according to your suggestions we have highly improved the quality of our paper.
